# Time to breastfeeding cessation and its predictors among mothers who have children aged two to three years in Gozamin district, Northwest Ethiopia: A retrospective follow-up study

**Tilahun Degu Tsega**[1]*, **Yilkal Tafere**[1], **Wassachew Ashebir**[2], **Biachew Asmare**[3]

**1** Epidemiology Unit, Department of Public Health, College of Health Sciences, Debre Markos University, Debre Markos, Ethiopia, **2** Department of Reproductive Health, College of Health Sciences, Debre Markos University, Debre Markos, Ethiopia, **3** Department of Human Nutrition, College of Health Sciences, Debre Markos University, Debre Markos, Ethiopia

* mkdt2121@gmail.com

## Abstract

### Introduction

Globally, breastfeeding duration is below the recommended level. In Ethiopia, more than 24% of mothers ceased breastfeeding before 24 months of age of a child which caused 14,000 preventable childhood deaths annually. To tackle this problem, current and up-to-date information regarding the time to breastfeeding cessation and its predictors is essential. Therefore, this study aims to determine the time to breastfeeding cessation and its predictors among mothers who have children aged two to three years in Gozamin district, Northwest Ethiopia.

### Methods

A community-based retrospective follow-up study was used among 502 mothers who have children aged two to three years in the Gozamin district from October 1, 2017, up to September 30, 2020. Interviewer-administered structured questionnaires were used. Cox proportional hazard model was applied after its assumptions and model fitness were checked, to identify predictors for time to breastfeeding cessation.

### Results

The overall mean time to breastfeeding cessation was 22.56 (95%CI: 22.21, 22.91) months, and the cumulative survival probability on breastfeeding up to 24 months was 82.5% (95% CI:78.85, 85.53). The overall incidence of early breastfeeding cessation was 7.77 (95% CI:6.31, 9.58) per 1000 person-month observations. Having no antenatal care follow up (AHR:2.15, 95%CI:1.19, 3.89), having ≥4 number of children (AHR:1.76, 95%CI:1.10, 2.80), < 24 months breastfeeding experience (AHR:1.77, 95%CI:1.14, 2.75), and presence

**Data Availability Statement:** All relevant data are within the manuscript and its Supporting Information files.

**Funding:** The author(s) received no specific funding for this work.

**Competing interests:** The authors have declared that no competing interests exist.

**Abbreviations:** AHR, Adjusted Hazard ratio; CHR, Crude Hazard ratio; CI, Confidence Interval; BFC, Breastfeeding Cessation; WHO, World Health Organization.

of cow milk in the household (AHR:3.01, 95%CI:1.89, 4.78) were significant predictors for time to breastfeeding cessation.

## Conclusion

The time to breastfeeding cessation is below the recommendation and therefore, strengthening breastfeeding education and related counseling at the community level is better.

## Introduction

Breastfeeding is a natural practice that is considered by almost all mothers at birth that gave breast milk for the healthiest start of the life of children [1, 2]. It promotes cognitive development, acts as a baby's first vaccine, providing critical protection from disease and death of a child and it also promotes maternal health by preventing breast cancer, ovarian cancer, and type II diabetes mellitus, which further lowers health care costs, and strengthening the development of nations by creating healthier families [2, 3].

World Health Organization (WHO) and United Nations Children Fund (UNICEF) recommended that children should breastfeed for six months exclusively and should continue up to the second birth date or above [4, 5]. Breast milk gives all nutritional requirements for six months of age and half of all nutritional requirements onwards up to one year of age [6, 7]. Breastfeeding from one up to two years of age also provides one-third of the nutritional requirements of the child [6].

Despite these benefits breastfeeding duration is below the recommended level. Worldwide more than one-third of children are ceased breastfeeding before two years among children aged 12–23 months [4]. In the poorest families also 36% of children ceased breastfeeding before their second year of birth [1]. Similarly, in southern and eastern Africa 29% of children were not benefiting from the recommendation of breastfeeding [4]. More than 24% of children in Ethiopia also ceased breast milk consumption contrary to the WHO recommendation [8].

Breastfeeding cessation before a recommended level could cause 823,000 deaths in under-five children and 20,000 mothers' death from breast cancer worldwide annually [9]. In sub-Saharan countries, 334,892 deaths of children occurred due to cessation of breastfeeding before WHO recommendation [10]. The risk of mortality due to infectious-related causes was also twofold higher among children who ceased breastfeeding when compared to breastfed children aged 6–23 months [11] and specifically, diarrheal-related morbidity and mortality was 2.18% in children who ceased breastfeeding during this period [12]. In Ethiopia also 14, 000 preventable childhood deaths and 5 million cases of diarrhea and pneumonia were occurred annually because of breastfeeding cessation before the recommended time [13].

Different sociodemographic related factors such as the age of the mother, marital status of the mother, educational status of the mother, occupation, wealth index, and the number of children were factors identified as predictors for time to breastfeeding cessation from previous works of literature [14–21]. Similarly, obstetric-related predictors, health service-related predictors, and nutrition-related predictors were also identified as factors for time to breastfeeding cessation [14, 22–27] (Fig 1).

Despite the Ethiopian government's effort on breastfeeding promotion and awareness creation, a significant number of children are still vulnerable to the effect of early breastfeeding cessation, and also the duration of breastfeeding is variable among regions with a decreasing trend over time [28]. Postpartum insusceptibility due to breastfeeding had an inhibiting effect

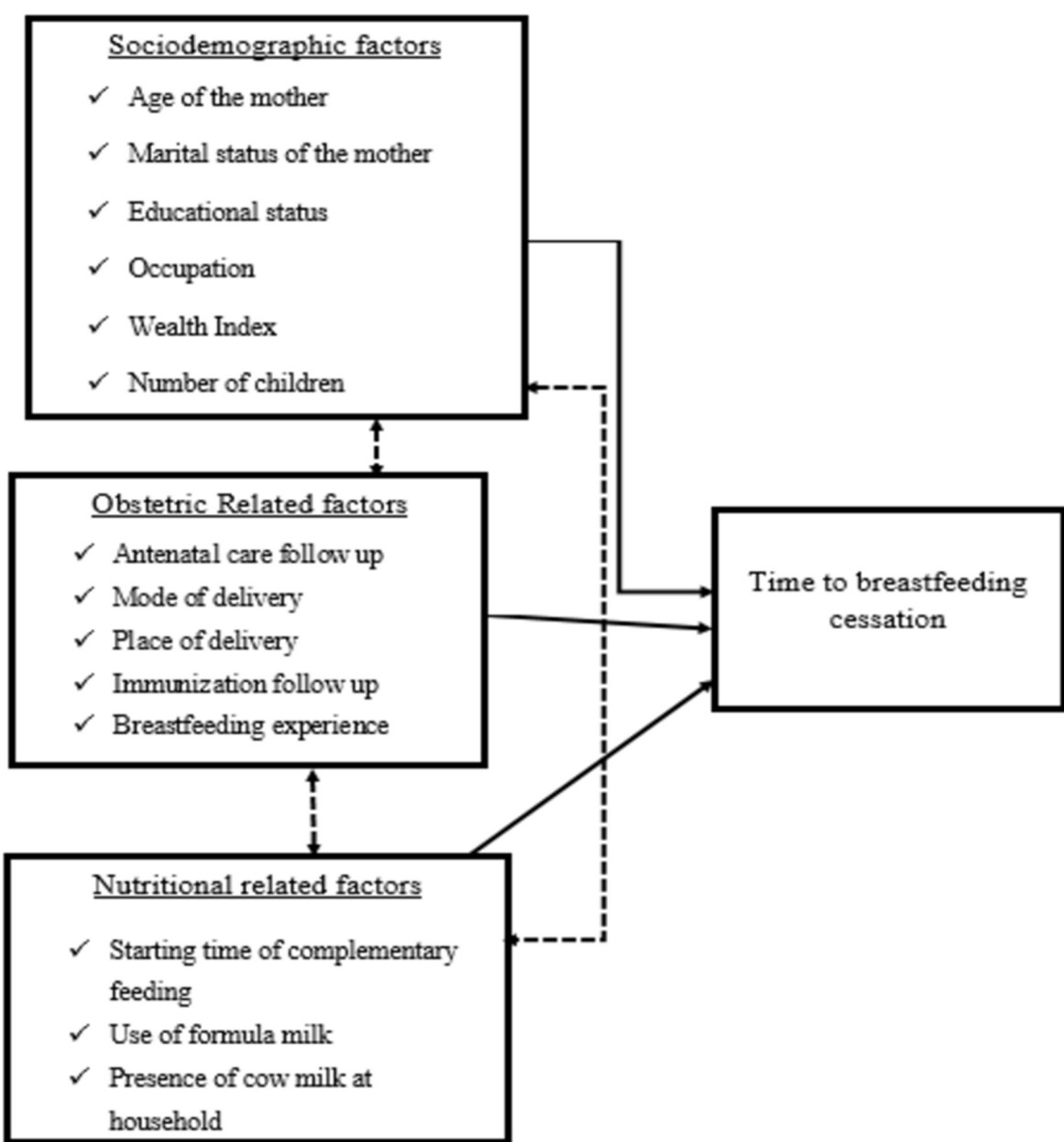

**Fig 1. Conceptual framework for time to breastfeeding cessation and its predictors among mothers who have children aged two to three years in Gozamin district, Northwest Ethiopia, 2020.**

on fertility among rural areas in the Amhara region of Ethiopia where breastfeeding duration was long [29], but the duration of breastfeeding is in a declined trend which was not searched well [28]. Previous studies on breastfeeding cessation (mostly exclusive breastfeeding cessation) were concentrated on the urban or at the institution level [22, 30]. Hence, there is limited data in this area especially in the rural part of Ethiopia, assessing the time to cessation of breastfeeding and its predictors in the rural part of Gozamin District is crucial for up-to-date information and scientific-based decision making and intervention. Therefore, this study assesses the time to breastfeeding cessation and its predictors among mothers who had children aged two to three years in Northwest Ethiopia of Gozamin District.

## Methods

### Study design

A community-based retrospective follow-up study design was used.

### Study area and period

This study was conducted at Gozamin district, which is located in East Gojjam Zone, Northwest Ethiopia, 299 km from Addis Ababa, the capital city of Ethiopia, and 265km from Bahir Dar, the regional capital city of Amhara. The District has 30 Kebeles (a Kebele is the lowest administrative level in Ethiopia) which has 22,316 under-five children, 12,872 under three years' children, 8,323 are under two years and 4,549 children are aged two to three years based on the Gozamin district health information system data and 2020 annual health office report (Gozamin district Health Office plan and performance annual report, 2020: 218.).

This study was conducted from October 1, 2017, to September 30, 2020, among mothers who have children aged two to three years, and data were collected from October 1, 2020, to November 30, .2020.

### Population

**Source population.** All mothers who have children aged two to three years in Gozamin District.

**Study population.** Mothers who have children aged two to three years living at selected Kebeles of Gozamin District during the data collection period were included in the study.

**Exclusion criteria.** Those mothers who were seriously ill and then become unable to communicate. Mothers who have a child aged two to three years not initially breastfed at least once were excluded. Mothers who gave care and breastfed other than their children were also excluded from this study.

### Sample size determination and procedures

**Sample size determination.** The sample size for this study was calculated using the general formula of sample size calculation for time to event data [31]:

$$\text{Sample size(n)} = \frac{\text{number of events(E)}}{\text{Probability of event (p(E))}}$$

$$\text{Where, Number of events (E)} = \frac{(za/2 + z_{1-B})^2}{P_1 P_2 (\ln HR)^2}$$

Probability of event(p(E)) = 1-(p₁s₁(t) + p₂s₂(t))

in which $z_{a/2}$ = 1.96 at 0.05 significant level, $z_{1-B}$ = 80% power, $p_1$ = proportion of population allocated to a non-exposure group, $p_2$ = proportion of population allocated to exposure group, non-response rate (W) = 10%, design effect = 2, survival probability ((s₁(t) and s₂(t) of a non-exposed group and exposed group respectively)) and Hazard ratio of predictors was taken from the literature [22]. Then Stata 14.1 software was used. Finally, the total sample size was 516 mother-child pairs.

**Sampling procedures.** The overall sampling procedure used was the multistage sampling technique. From thirty kebeles in the district, ten Kebeles were selected by simple random sampling technique having a total of 1673 eligible mother-child pairs, out of which 516 eligible participants were selected. Then the eligible participants were allocated for each kebele using the population proportion formula. Using a systematic random sampling technique, mother-child pairs were selected with a sampling interval of three eligible households. Finally, to obtain mother-child pairs needed for this study, data collectors moved every household within the selected kebele.

## Variables of the study

**Dependent variable.** Time to breastfeeding cessation.
**Independent variables.**

➢ Age of mother

➢ Marital status of the mother

➢ The educational level of the mother

➢ Occupation

➢ Wealth index

➢ Number of children

➢ Antenatal care follow-up

➢ Place of delivery

➢ Mode of Delivery

➢ Immunization follow-up

➢ Breastfeeding experience

➢ Starting time of complementary feeding

➢ Formula milk use status

➢ Presence of cow milk at household

## Operational definitions

**Breastfeeding cessation.** Breastfeeding was stopped before 24 months of age of a child.
**Event.** Stopped breastfeeding before 24 months of age of a child as reported by the mother during the data collection period.
**Censored.** A mother who was breastfed the child during the data collection period or ceased after two years.
**Survival time.** The time from initiation of breastfeeding (birth) until the cessation or censoring of breastfeeding in months.
**Wealth index.** A score was given to each household based on the relative ownership of assets of the house. These scores were derived using principal component analysis. Finally, the household's wealth has been categorized as poor (1), middle (2), and rich (3) based on the wealth status score [28, 32].

## Data collection tools and procedures

A structured questionnaire was developed and adapted from works of literature written in the Ethiopian context [22, 28]. The questionnaire contains maternal sociodemographic, obstetric, health service, and nutritional-related predictors.

Data were collected using smartphone-assisted interviewer-administered questionnaires through EPI-INFO android version 7.2 software. For data collectors, supervisors, and the principal investigator, EPI-INFO version 7.2 was installed on their smartphones and then the template of the questionnaire was loaded. Data were collected at a time when mothers were easily accessed at home such as weekends or holidays of the community through assisting guiders. The survival data has been collected from mothers who had children aged two to three years.

Mothers were asked to answer the date of breastfeeding cessation, and the child's birth date was the beginning point of the retrospective follow up study, and the endpoint of the follow-up study was taken as the date of breastfeeding cessation or the end of the study, which was the length of the survival time. The event of interest for this study was mothers who had stopped breastfeeding before 24 months of the child, and those breastfed during data collection or stopped after 24 months were right-censored. Mothers who were not presented during the data collection time were further considered the next two holy or weekend days and then after, the mother was considered as non-response mother-child pairs.

## Data quality assurance

To maintain data quality, the principal investigator trained four data collectors and two super-visors for two days. A pretest was conducted on 26 mother-child pairs (5%) from a non-selected kebele. After the pretest, part of the questionnaire which was not easily understood by the respondents was modified. Some variables missed during the pretest were added to the final questionnaire. On-site supervision was performed and each synced file of the question-naire was sent by each data collector on daily basis to the email of the principal investigator for completeness and accuracy checking before leaving the study area.

## Data processing and analysis

Data were collected and entered into EPI-INFO version 7.2 and exported to Microsoft office excel 2016, and then further exported to Stata 14.1 for further coding, cleaning, and analysis. Checking for missing values and, the presence of influential outliers was evaluated.

Months were used as a time scale to measure the time to breastfeeding cessation. Each participant's outcome was dichotomized into the event and censored coded as "1" and "0" respectively. Descriptive statistics such as mean (standard deviation) for normally distributed data, median (interquartile range), frequencies, and proportions were used to describe the characteristics of the study participants. To estimate the breastfeeding survival status of the mother Kaplan-Meier survival curve and the log-rank test was used. A life table was also used to estimate the cumulative survival probability of mothers breastfeeding up to 24 months. The incidence rate of early cessation of breastfeeding of mothers was calculated as the number of events over the person-months of follow-up.

The Cox Proportional Hazard (PH) assumptions were checked using Schoenfeld residuals statistical test, presence of time-dependent covariate, and graphical methods. The model adequacy was also checked using the Cox-Snell residuals graph and Schoenfeld residuals statistical global test.

Bi-variable Cox regression model building was done for each independent variable and outcome of interest to identify potentially significant variables with a significant test $\leq 0.2$ for the multivariable Cox proportional hazards regression model. Then, multivariable analysis was started with a model containing all of the selected variables, and then a stepwise backward regression procedure was applied.

Hazard ratio (HR) with 95% confidence intervals (CI) was used to interpret the result of the final model. Statistical significance was declared at the p-value is less than 0.05. Finally, the result of this study was presented using tables, graphs, or text narrations.

## Ethical procedures

Ethical clearance was obtained from the ethical review committee of Debre Markos University, College of Health Sciences (Ref. No: HSC/R/C/Ser/Co/43/11/13) for the Gozamin district Health office, and permission was obtained. All information collected from mothers was kept

strictly confidential by excluding Personal identifiers from the questionnaire and codes were used. Oral informed consent and then finger stamped signature from the mothers were obtained.

## Result

Out of 516 mother-child pairs that were entered into the study, 502 mother-child pairs were responded at the data collection period and gave a response rate of 97.3%.

### Socio-economic and demographic characteristics

The mean baseline age of mothers was found to be 28.4 ± 6.08 SD years. Among the total 502 mother-child pairs, 449 (89.44%) were married and 368 (73.31%) have no formal educational level. The median age of children was 29 (IQR = 26–33) months. Among 16 never-married mothers, 7 were ceased breastfeeding their children early during the follow-up time of this study. Likewise, among mothers who had four and fewer children in their household, 60 (15.38%) mothers have ceased breastfeeding before 24 months (Table 1).

### Obstetric and health service utilization characteristics of respondents

Out of the total mothers enrolled in the study, 457 (91.04%) had ANC follow-up. Among mothers who ceased breastfeeding early, 71 (80.68%) had ANC follow up of which 24 and 47 of them had <4 and ≥4 number of visits respectively (Table 2).

### Breastfeeding and nutritional related characteristics of respondents

Among the total observations, 382 (76.10%) of mothers had breastfeeding experience and of these 54 (14.14%) had below 24 months of experience. Three hundred five (60.76%) of

**Table 1. Socio-economic and demographic characteristics of mother-child pairs in Gozamin district, Northwest Ethiopia from October 1/2017-September 30/2020.**

| Variable | Category | Number(%) | BFC Status[**] | |
|---|---|---|---|---|
| | | | Event (%) | Censored(%) |
| Maternal age | 15–24 | 133 (26.49) | 26 (29.55) | 107 (25.85) |
| | 25–34 | 285 (56.77) | 42 (47.73) | 243 (58.70) |
| | 35 and above | 84 (16.73) | 20 (22.73) | 64 (15.46) |
| Marital status of the mother | Married | 449 (89.04) | 76 (86.36) | 373 (90.10) |
| | Never married | 16 (3.19) | 7 (7.95) | 9 (2.17) |
| | Widowed/Divorced | 37 (7.37) | 5 (5.68) | 32 (7.73) |
| Educational level of the mother | No Education | 368 (73.31) | 63 (71.59) | 305 (73.67) |
| | Primary education | 72 (14.34) | 11 (12.50) | 61 (14.73) |
| | 2ndary education and Higher | 62 (12.35) | 14 (15.91) | 48 (11.59) |
| Occupation of the mother | Housewife | 404 (80.48) | 71 (80.68) | 333 (80.43) |
| | Employed | 48 (9.56) | 6 (6.82) | 42 (10.14) |
| | Daily Laborer | 26 (5.18) | 5 (5.68) | 21 (5.07) |
| | Others* | 24 (4.78) | 6 (6.82) | 18 (4.35) |
| Number of children | ≤4 | 390 (77.69) | 60 (68.18) | 330 (79.71) |
| | >4 | 112 (22.31) | 28 (31.82) | 84 (20.29) |
| Wealth index | Poor | 168 (33.47) | 30 (34.09) | 138 (33.33) |
| | Middle | 220 (43.82) | 38 (43.18) | 182 (43.96) |
| | Rich | 114 (22.71) | 20 (22.73) | 94 (22.71) |

Others* include Tea/coffee/local beer sellers, students BFC status** = breastfeeding cessation.

**Table 2. Obstetric, breastfeeding, nutrition-related factors and health service utilization characteristics of mother-child pairs in Gozamin district, Northwest Ethiopia from October 1/2017-September 30/2020.**

| Variable | Category | Number(%) | BFC status** | |
|---|---|---|---|---|
| | | | Event (%) | Censored(%) |
| Antenatal care follow up | No | 45 (8.96) | 17 (19.32) | 28 (6.76) |
| | < 4 | 133 (26.49) | 24 (27.27) | 109 (26.33) |
| | ≥4 | 324 (64.54) | 47 (53.41) | 277 (66.91) |
| Place of delivery | Home | 132 (26.29) | 27 (30.68) | 105 (25.36) |
| | Health Institution | 370 (73.71) | 61 (69.32) | 309 (74.64) |
| Mode of delivery | Vaginal delivery | 460 (91.63) | 78 (88.64) | 382 (92.27) |
| | Cesarean section | 42 (8.37) | 10 (11.36) | 32 (7.73) |
| Immunization follow up status | Yes | 444 (88.45) | 69 (78.41) | 375 (90.58) |
| | Interrupted | 36 (7.17) | 11 (12.50) | 25 (6.04) |
| | No | 22 (4.38) | 8 (9.09) | 14 (3.38) |
| Breastfeeding Experience | Yes | 382 (76.10) | 69 (78.41) | 313 (75.60) |
| | No | 120 (23.90) | 19 (21.59) | 101 (24.40) |
| Starting time of complementary feeding | < 6 months | 30 (5.98) | 12 (13.64) | 18 (4.35) |
| | = 6 months | 305 (60.76) | 52 (59.09) | 253 (61.11) |
| | >6 months | 167 (33.27) | 24 (27.27) | 143 (34.54) |
| Use of formula milk status | Yes | 40 (7.97) | 9 (10.23) | 31 (7.49) |
| | No | 462 (92.03) | 79 (89.77) | 383 (92.51) |
| Presence of cow milk at HH* | Yes | 62 (12.35) | 26 (29.55) | 36 (8.70) |
| | No | 440 (87.65) | 62 (70.45) | 378 (91.30) |

BFC status** = breastfeeding cessation HH* = Household.

participants had started at the right time of complementary feeding initiation for their current child, but 30 (5.98%) and 167 (33.67%) had started complementary feeding before and after six months of age respectively (Table 2).

## Survival status of mothers on breastfeeding

Among 502 observations followed, 414 (82.47%) observations were censored at the end of the study. The minimum and maximum follow-up time of the cohort was 2 months and 24 months respectively. The median follow-up time of the cohort was 24 months.

From the life table, the cumulative survival probability of breastfeeding up to 12, and 24 months were 0.968, and 0.825 respectively. Therefore, the proportion of mothers who breastfed the current child until 24 months was 82.5% (95%CI: 78.85%, 85.53%) (Table 3).

The overall survival time on breastfeeding was also estimated by the Kaplan-Meier survival curve, which indicated the slow occurrence of events over the follow-up period (Fig 2). The survival estimates on breastfeeding were varied with antenatal care follow-up, starting time of

**Table 3. Cumulative failure probability of breastfeeding among mothers who have children aged two to three years in Gozamin district from October 1/2017-September 30/2030.**

| Time interval (Months) | Population entered into the interval | Event | Censored | Cumulative failure probability | 95% Confidence interval |
|---|---|---|---|---|---|
| 6 | 502 | 7 | 0 | 0.0139 | 0.0067, 0.0290 |
| 12 | 495 | 9 | 0 | 0.0319 | 0.0196, 0.0515 |
| 18 | 486 | 25 | 0 | 0.0817 | 0.0608, 0.1093 |
| 24 | 461 | 47 | 414 | 0.1753 | 0.1447, 0.2115 |

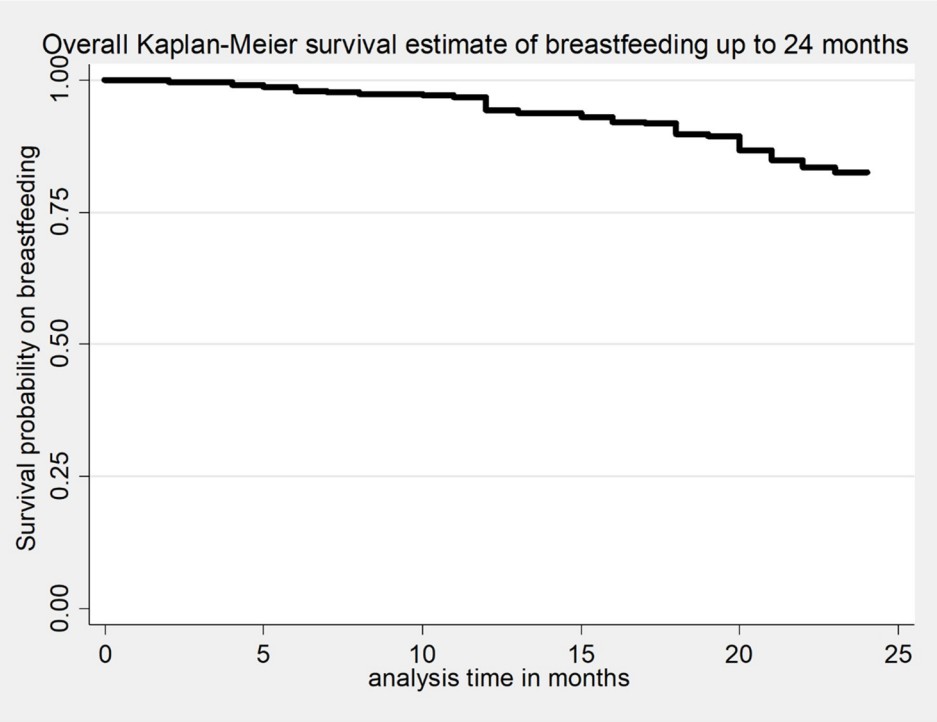

**Fig 2. The Kaplan-Meier survival estimates of time to breastfeeding cessation among mothers who have children aged two to three years in Gozamin district, Northwest Ethiopia from October 1/2017-September 30/2020.**

complementary feeding, and presence of cow milk in the household. This survival status was compared and tested significant statistically using a log-rank test (Figs 3–5).

The mean time to cessation of breastfeeding was 22.56 months (95% CI: 22.21, 22.91). Among those who ceased breastfeeding early, the median time to cessation of breastfeeding was 18 (IQR = 12–20, 95%CI: 15, 19) months.

## Incidence of early breastfeeding cessation

Out of the total observations, 88 (17.53%) mothers have ceased breastfeeding before 24 months with an overall incidence rate of 7.77 per 1000 (95% CI: 6.31, 9.58) person-month observations after 11325-lifetime risk follow-up months.

From the total events, the highest incidence of breastfeeding cessation was observed at end of the 18th and 24th months of follow-up with the incidence rate of 8.21 (95%CI:5.45, 12.35) and 13.88 (95%CI: 10.01, 19.25) per 1000 person-months observations respectively. Besides the incidence of breastfeeding cessation before 6 and 12 months of follow-up were 3.67 (95% CI: 2.03, 6.62) and 6.14 (95%CI: 3.87, 9.74) per 1000 person-month observations respectively. Likewise, the incidence rate of early breastfeeding cessation among mothers who had no ante-natal care follow-up and had cow milk in the household was 17.95 (95%CI:11.16, 28.88) and 20.34 (95%CI:13.85, 29.88) per 1000 person-months observations respectively.

## Predictors of time to breastfeeding cessation

Predictors that had an association at p-value ≤0.2 in bi-variable cox proportional hazard regression were included in multivariable regression. Accordingly, age of the mother, marital

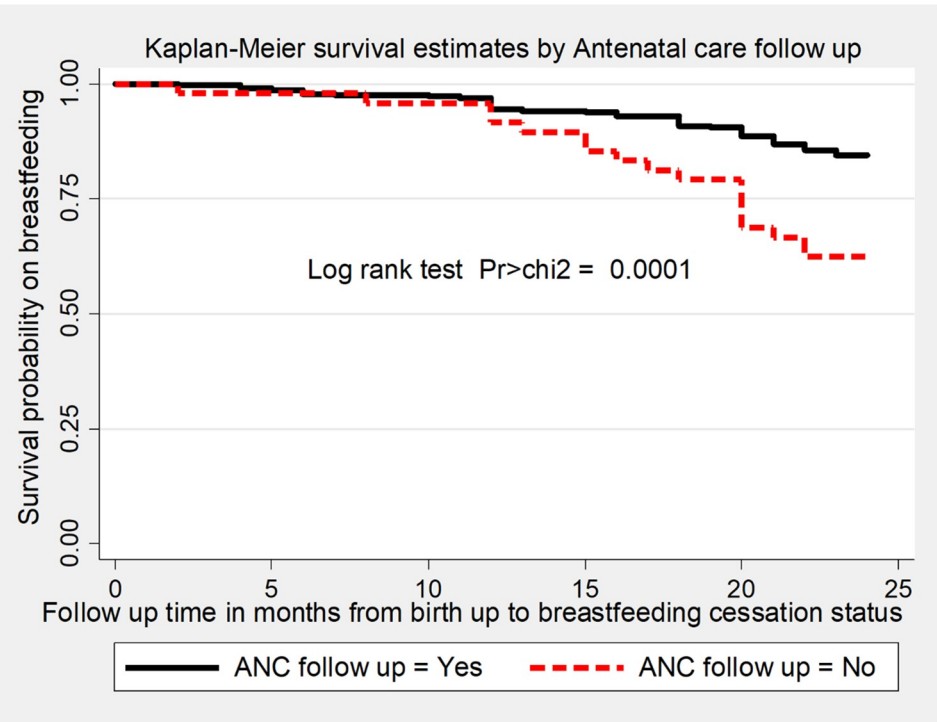

**Fig 3. The Kaplan-Meier survival estimates graph by antenatal care follow up of the mother for time to breastfeeding cessation among mothers who have children aged two to three years in Gozamin district, Northwest Ethiopia from October 1/2017-September 30/2020.**

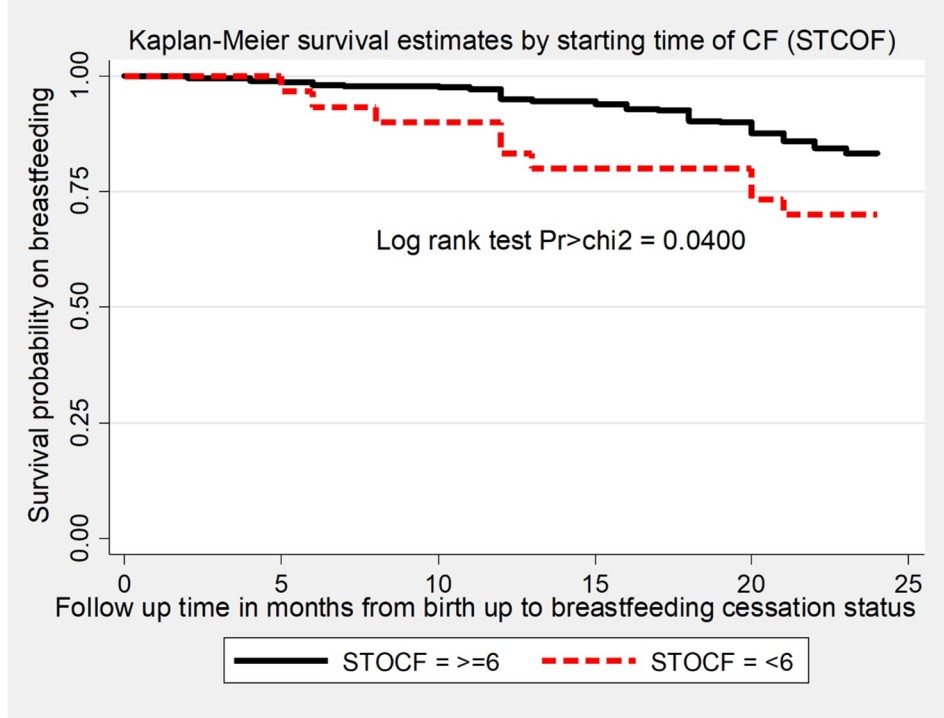

**Fig 4. The Kaplan-Meier survival estimates graph by starting time of complementary feeding of the mother for time to breastfeeding cessation among mothers who have children aged two to three years in Gozamin district, Northwest Ethiopia from October 1/2017-September 30/2020.**

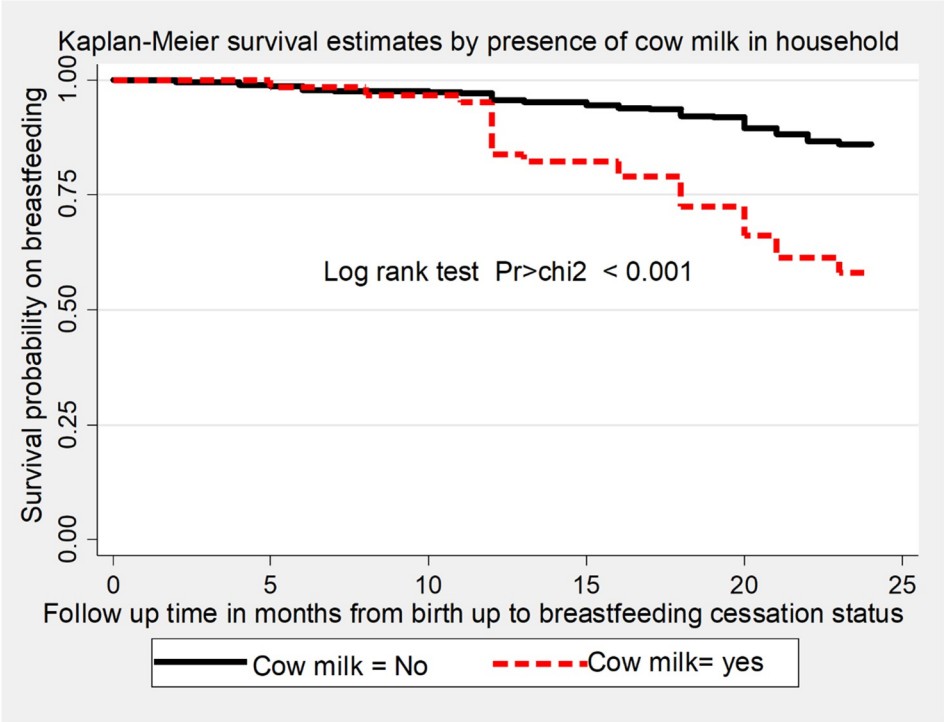

**Fig 5. The Kaplan-Meier survival estimates graph by presence of cow milk at household of the mother for time to breastfeeding cessation among mothers who have children aged two to three years in Gozamin district, Northwest Ethiopia from October 1/2017-September 30/2020.**

status of the mother, educational status of the mother, number of children, antenatal care follow-up, mode of delivery, immunization follow-up, breastfeeding experience, starting time of complementary feeding, and presence of cow milk at household were included into multivariable cox proportional hazard regression model. Finally, after the stepwise backward elimination approach four variables such as >4 number of children, having no antenatal care follow up, having <24 months breastfeeding experience and presence of cow milk at household were statistically significant predictors of time to breastfeeding cessation at p-value <0.05 level of significance (Table 4).

Having greater than four children at household was a statistically independent sociodemographic predictor of time to breastfeeding cessation in which mothers who had ≥4 number of children were 1.76 times (AHR = 1.76, 95%CI:1.11,2.80) higher to ceased breastfeeding early than mothers who had ≤4 number of children at household. The risk of early breastfeeding cessation was 2.15 times higher among mothers who had no antenatal care follow-up than mothers who had ≥4 antenatal care follow up (AHR = 2.15, 95% CI:1.19, 3.89). Similarly, the risk of early breastfeeding cessation was 1.77 times higher among mothers who had less than 24 months of breastfeeding experience than mothers who had ≥24 months breastfeeding experience (AHR = 1.77, 95%CI:1.14, 2.75). The presence of cow milk in the household was also a statistically independent predictor of time to breastfeeding cessation in which the risk of early breastfeeding cessation was 3.01 times higher than mothers who had no cow milk at their household (AHR = 3.01, 95%CI:1.89, 4.78) (Table 4).

**Table 4. Bi-variable and multivariable cox proportional hazard regression analysis of predictors of time to breast-feeding cessation among mothers who had children aged two to three years in Gozamin district from October 1, 2017-September 30/2020.**

| S.no | Variables | CHR (95%CI) | AHR (95%CI) | P-value |
|------|-----------|-------------|-------------|---------|
| 1 | Number of children | | | |
| | ≤ 4 | 1 | 1 | |
| | >4 | 1.70 (1.09, 2.68) | 1.76 (1.10, 2.80) | 0.018 |
| 2 | Antenatal care visit | | | |
| | ≥4 | 1 | 1 | |
| | <4 | 1.30 (0.79, 2.12) | 1.24 (0.76, 2.04) | 0.391 |
| | No | 2.96 (1.70, 5.16) | 2.15 (1.19, 3.89) | 0.011 |
| 3 | Breastfeeding experience | | | |
| | ≥24 months | 1 | 1 | |
| | <24 months | 1.8 (1.23, 2.84) | 1.77 (1.14, 2.75) | 0.011 |
| 4 | Immunization follow-up | | | |
| | Fully | 1 | 1 | |
| | Not fully | 2.26 (1.36, 3.75) | 1.50 (0.87, 2.57) | 0.143 |
| 5 | Presence of cow milk | | | |
| | No | 1 | 1 | |
| | Yes | 3.49 (2.21, 5.52) | 3.01 (1.89, 4.78) | <0.001 |

## Discussion

Cessation of breastfeeding before the recommended time increases the risk of developing diarrhea, respiratory infectious disease, and death of a child, which further affects child survival and morbidity [33, 34]. In Ethiopia, more than 24% of children are vulnerable to the effect of breastfeeding cessation [8]. Therefore, this study assesses the incidence rate and the mean time to breastfeeding cessation, and its predictors among mothers who have children aged two to three years in the Gozamin district.

The overall incidence rate of breastfeeding cessation before 24 months of age of the child was 7.77 per 1000 person-month observations. This finding is lower than the finding of the study conducted in Debre Markos town which reported that the overall incidence rate of breastfeeding cessation before 24 months of age of the child was 13.70 per 1000 person-month observations [22]. It might be due to socioeconomic differences between the urban and rural study populations of the two studies. Similarly, the finding of the study in Iran, Tehran was 16.02 per 1000 person-months observation, which is higher than the finding of the present study [14]. The reason may be the present study is a community-based retrospective follow-up study which may have a relatively high recall bias than register-based follow-up studies. This may underestimate the incidence rate of breastfeeding cessation before 24 months of age of the child.

The finding of this study stated that the proportion of mothers who were breastfeeding up to the 24[th] month of follow-up was 82.5%. This finding is higher than the finding of the study done in Debre Markos town, which indicated that the proportion of mothers breastfeeding until the 24[th] month of the life of the child was 68.5% [22]. The reason might be the finding of the current study was based on rural mother-child pairs which were mainly less-educated mothers who tended to breastfeed for a longer time. Likewise, the proportion of mothers to breastfeed their children up to 24[th] months of age was 72.1% done in Ethiopia at the national level, which is less than the finding of the current study [8]. It might be due to the

sociodemographic difference of the study population by the national study and it also incorporated urban mothers intended to cease early than the rural mothers.

The overall estimated mean time to breastfeeding cessation was 22.56 months, which means that as mothers' followed for 24 months, the survival time on breastfeeding of the current child was averagely estimated as 22.56 months. This finding is higher than the finding of the study done in Iran, Tehran, which is 21.49 months [14]. This might be due to the present study being done on rural mothers who might breastfeed for a longer duration than urban mothers. As urban mothers are more educated than rural mothers, the probability of engaging in full-time work is higher than rural mothers which leads to early cessation of breastfeeding.

The risk of early breastfeeding cessation was higher among mothers who had More than 4 children as compared to mothers who had ≤4 children. This finding is supported by the finding of the study done in Debre Markos town, Ethiopia, which stated that mothers having a higher number of children were at higher risk of early breastfeeding cessation than mothers having a lower number of children [22]. The possible justification might be as the number of children in the household increases the workload of the mother for caring for children is increased. The time for breastfeeding the current child decreases gradually which makes the mother to ceased breastfeeding early.

The risk of early breastfeeding cessation was higher among mothers who had no antenatal care follow-up as compared to mothers who had four and above antenatal care follow-up. The finding of this study is opposite to the finding of the study done in Bangladesh [25]. The difference might be in Bangladesh urban mothers received more antenatal care than rural mothers who breastfed shorter duration [35]. This finding is supported by the finding of the national study in Ethiopia reported that mothers who didn't receive antenatal care services had a 10% shorter duration of breastfeeding than mothers who received 4$^+$ antenatal care services [32]. This might be as a mother receives more antenatal care service, counseling regarding breastfeeding, and its duration was increased, which further increases the knowledge of a mother on the advantages of longer duration of breastfeeding.

Breastfeeding experience less than 24 months was a risk factor for early breastfeeding cessation. This finding is supported with the finding of the study done in China, Wuhan, which stated that short previous breastfeeding duration and negative previous breastfeeding experience have an unfavorable effect on subsequent breastfeeding duration [27]. The reason might be longer breastfeeding experienced mothers' had better breastfeeding attitudes, confidence, self-efficacy, motivation, and intention for a longer duration of breastfeeding [36].

The presence of cow milk in the household during the last 24 months is an independent statistically significant predictor of time to breastfeeding cessation in this study. Mothers who had cow milk in the household have a higher risk of early breastfeeding cessation. This finding is consistent with the study done in southern Ethiopia [23]. The reason behind this may be mothers who have cow milk at their own home may give it as a substituent for breast milk starting from early life that may lead to early cessation of breastfeeding. This is evidenced in which mothers gave cow milk even before six months of age of the child [37].

## Limitation of the study

It is a retrospective study, there might be recall bias, possibly resulting in over or underestimation of actual time to breastfeeding cessation. The survival time of breastfeeding duration for this study is determined using a restricted mean, which may underestimate the overall time to breastfeeding cessation. Due to the retrospective nature of the study, some variables such as the perception of sufficient milk supply, knowledge, and attitude of mothers were not assessed.

## Conclusion

This study assessed the incidence rate and time to breastfeeding cessation. The incidence rate of early breastfeeding cessation was 7.77 per 1000-person month observations which is lower than previous works of literature. The mean time to breastfeeding cessation was 22.56 months which is below the WHO recommendation. Having >4 number of children, having no antenatal care follow-up, <24 months breastfeeding experience and presence of cow milk were statistically significant predictors of time to breastfeeding cessation. However, immunization follow-up and <4 number of antenatal care follow-up were not statistically significant predictors.

Therefore, stakeholders better to take measures further to prolong the duration of breastfeeding through incorporating these significant predictors as one part of breastfeeding counseling and education at the community level.

## Supporting information

**S1 Dataset.**
(DTA)

**S1 Questionnaire.**
(ZIP)

## Acknowledgments

We would like to thank Debre Markos University and Gozamin district health office staffs for their cooperation. Further, we would like to thank study participants, data collectors, and community guiders.

## Author Contributions

**Conceptualization:** Tilahun Degu Tsega, Yilkal Tafere, Wassachew Ashebir, Biachew Asmare.

**Data curation:** Tilahun Degu Tsega, Yilkal Tafere, Wassachew Ashebir, Biachew Asmare.

**Formal analysis:** Tilahun Degu Tsega.

**Investigation:** Tilahun Degu Tsega.

**Methodology:** Tilahun Degu Tsega, Yilkal Tafere, Wassachew Ashebir.

**Software:** Tilahun Degu Tsega, Yilkal Tafere, Wassachew Ashebir.

**Supervision:** Yilkal Tafere, Wassachew Ashebir, Biachew Asmare.

**Validation:** Tilahun Degu Tsega, Yilkal Tafere, Wassachew Ashebir, Biachew Asmare.

**Visualization:** Tilahun Degu Tsega.

**Writing – original draft:** Tilahun Degu Tsega.

**Writing – review & editing:** Yilkal Tafere, Wassachew Ashebir, Biachew Asmare.

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
