## [Decision Letter · Decision Letter 0]

18 Oct 2021

PONE-D-21-09209Time to early breastfeeding cessation and its predictors among mothers who have children aged two to three years in Gozamin district, Northwest Ethiopia: a retrospective follow-up studyPLOS ONE

Dear Tilahun Tsega Degu,

Thank you for submitting your manuscript to PLOS ONE. After careful consideration, we feel that it has merit but does not fully meet PLOS ONE’s publication criteria as it currently stands. Therefore, we invite you to submit a revised version of the manuscript that addresses the points raised during the review process.

We look forward to receiving your revised manuscript.

Kind regards,

Gouranga Lal Dasvarma, PhD

Academic Editor

PLOS ONE

Additional Editor Comments (if provided):

Dear Tilahun Tsega Degu,

Please address the comments and suggestions of the two reviewers satisfactorily. Please also make sure that the revised manuscript follows PLOS One guidelines about formatting and referencing and it is edited for English language.

Journal Requirements:

2. Thank you for including your ethics statement:  "Debre Markos University, College of Health Sciences Ref. No: HSC/R/C/Ser/Co/43/11/13".   

Please amend your current ethics statement to confirm that your named institutional review board or ethics committee specifically approved this study. 

3. Please provide additional details regarding participant consent. In the ethics statement in the Methods and online submission information, please ensure that you have specified:

 - whether consent was obtained

 - whether consent was informed 

 - what type of consent you obtained (for instance, written or verbal, and if verbal, how it was documented and witnessed). 

 - if your study included minors, state whether you obtained consent from parents or guardians. 

 - if the need for consent was waived by the ethics committee, please include this information.

4. Please include additional information regarding the survey or questionnaire used in the study and ensure that you have provided sufficient details that others could replicate the analyses. For instance, if you developed the survey or questionnaire as part of this study and it is not under a copyright more restrictive than CC-BY, please include a copy, in both the original language and English, as Supporting Information. If the questionnaire is published, please provide a citation to the (1) questionnaire and/or (2) original publication associated with the questionnaire.

6. We note that you have referenced "GDHO. Gozamin district Health Office plan and performance annual report, 2012 E.C.pdf. In: office DH, editor. annual ed. Unpublished ppt2020. p. 218" which has currently not yet been accepted for publication. Please remove this from your References and amend this to state in the body of your manuscript: "GDHO. Gozamin district Health Office plan and performance annual report, 2012 E.C.pdf. In: office DH, editor. annual ed. Unpublished ppt2020. p. 218" as detailed online in our guide for authors

http://journals.plos.org/plosone/s/submission-guidelines#loc-reference-style.

7. We note that you have indicated that data from this study are available upon request. PLOS only allows data to be available upon request if there are legal or ethical restrictions on sharing data publicly. For more information on unacceptable data access restrictions, please see http://journals.plos.org/plosone/s/data-availability#loc-unacceptable-data-access-restrictions. 

Reviewers' comments:

Reviewer's Responses to Questions

**Comments to the Author**

1. Is the manuscript technically sound, and do the data support the conclusions?

Reviewer #1: Partly

Reviewer #2: Partly

2. Has the statistical analysis been performed appropriately and rigorously? 

Reviewer #1: I Don't Know

Reviewer #2: I Don't Know

3. Have the authors made all data underlying the findings in their manuscript fully available?

Reviewer #1: Yes

Reviewer #2: Yes

4. Is the manuscript presented in an intelligible fashion and written in standard English?

Reviewer #1: Yes

Reviewer #2: No

5. Review Comments to the Author

Reviewer #1: Major comments

Title

You are using two terms (time and early) with similar meaning. I would suggest removing either of them.

The title is not clearly stated. I suggest: Early breastfeeding cessation and its predictors among mothers with two to three years old children in Gozamin....

Corresponding author

No need to mention these details here, put the adress next to your address above. To show the coresponding author, put * next to the name then mention what the * means. * Corresponding author

Abstract: What is the importance of mentioning the last sentence in the introduction? Delete it and add the aim/objective of your study

Methods (page 5)

I do not think that the you have used proper study design

You have explained too much about study area which is less important, try to summarize in few sentences.

What is your justification to take this timeframe October 2017 to September 20202?

The inclusion criteria is similar with the study population, try to summarize the subheadings and avoid repetition.

Exclusion criteria (last two lines), Not clear what you want to say? revise it

Sampling procedure: What is the total number of Kebele, from how much did you select ten?

Why do you use the word "infant chaild pairs" Have you collect any data from the child? I would suggest to use mothers....

data collection: I do not know why you have taken date of birth as starting point? it is obvious.

Your dependent variable is early BF cessation before 24 months and the participants are mothers with children 2-3 yeaars. So why are you using retrospective follow up design? which is used to assess the outcome of certain exposure. I think the design you have used is not correct. I am ready to hear your explanation about this.

The last three lines of page 8: To answer this, why do you use retrospective follow up? because it is obvious that they have started after birth then stopped in one specific time.

Data collection tool: This should come before data collection. I would suggest to merge with the above sub headings.

What was the findings of your pre test, summarize in one or two sentences. was there any problem with the tool? what do you find from your pretest?

Results

First sub title: Why are you referring as baseline? it is confusing. do you have any data taken during follow up? I think such category is used for multiphase study but yours is single phase study. Therefore, better to sat Sociodemographic characteristics

Fourth line of first sub heading: Did you had any follow up time in your study? I this the main problem is related to the study design.

What is the importance of mentioning this as incidence?

Minor comments

Title page: Be consistent, you have added affiliation to 2 and 3 but 1 and 4 did not have affiliation.

Put each email next to their address

Abstract: add comma to 14000, in the conclusion you have used different line spacing

data collection procedure there is a word "For 4 data collectors.... What? not clear

There are grammatical errors, the language need revision

Reviewer #2: The draft manuscript should convince clearly with evidence the breastfeeding cessation is a critical issue in Ethiopia. Otherwise, the analyses objective should be revised to be appropriated with the country evidence-based breastfeeding issue. There is a need for the study to develop a theory of what factors and their mechanisms to predicting the breastfeeding behaviour. The arguments provided in the background is often inconsistent (See my attached written review).

6. PLOS authors have the option to publish the peer review history of their article (what does this mean?). If published, this will include your full peer review and any attached files.

Reviewer #1: No

Reviewer #2: No

---

## [Author Response · Author response to Decision Letter 0]

25 Oct 2021

Response to Reviewers

Author Feedback: Dear editors and reviewers of this manuscript, we have no sufficient words to give our gratitude for your devotion of time and energy to improve the manuscript sufficient enough to be ready for production and then publication. As per your comments, we have tried to improve the manuscript and gave responses to those comments one by one. Dear editors and reviewers, even further we are ready to hear your comments and improve accordingly. Thanks a lot.

Reviewers' comments:

Reviewer's Responses to Questions

Comments to the Author

1. Is the manuscript technically sound, and do the data support the conclusions?

Reviewer #1: Partly

Reviewer #2: Partly

Author Response: yes, it is sounded technically as all research procedures were implemented accordingly. To assure this the data set which is in Stata format (.DAT) was attached as a supportive file. 

2. Has the statistical analysis been performed appropriately and rigorously?

Reviewer #1: I Don't Know

Reviewer #2: I Don't Know

Author response: yes, the appropriate and robust way to analyze time to event type of data was Cox regression, which is a popular and semiparametric type of modeling that nearly gives results obtained by parametric type of modeling in survival analysis at which data distribution was not well known. Hence, “time to breastfeeding cessation” is an outcome variable having a time-to-event nature that is handled by cox regression modeling with the assumptions of modeling were fulfilled. So the statistical analysis was performed using this modeling technique, it is appropriate.

3. Have the authors made all data underlying the findings in their manuscript fully available?

The PLOS Data policy requires authors to make all data underlying the findings described in their manuscript fully available without restriction, with rare exceptions (please refer to the Data Availability Statement in the manuscript PDF file). The data should be provided as part of the manuscript or its supporting information, or deposited to a public repository. For example, in addition to summary statistics, the data points behind means, medians, and variance measures should be available. If there are restrictions on publicly sharing data—e.g. participant privacy or use of data from a third party—those must be specified.

Reviewer #1: Yes

Reviewer #2: Yes

Author response: Yes, all data used for this manuscript were obtained without any restriction and provided as a “supportive file” in the web application.

4. Is the manuscript presented in an intelligible fashion and written in standard English?

Reviewer #1: Yes

Reviewer #2: No

Author response: Thank you, we accept your comments. Typographical and grammatical errors in the revised manuscript were corrected and made to be clear.

5. Review Comments to the Author

Reviewer #1: Major comments

Title

You are using two terms (time and early) with similar meanings. I would suggest removing either of them. The title is not clearly stated. I suggest: Early breastfeeding cessation and its predictors among mothers with two to three years old children in Gozamin....

Author response: Accepted and modified as the time to breastfeeding cessation and its predictors among mothers who have children aged two to three years in Gozamin district, Northwest Ethiopia. As we all know the title should be precise, predictive, and well explanatory, which means that a title should have a minimum word that at least tell the nature of the outcome variable, the type of statistical model, and the design that will use. If we made the title “early breastfeeding cessation and its predictors……”, it tells us logistic regression and cross-sectional study design were used. Hence it was modified as “time to breastfeeding cessation and its predictors……”. In this case, it indicates that the outcome variable is “time to event” and hence the statistical modeling used was “survival analysis”.

Corresponding author: No need to mention these details here, put the adress next to your address above. To show the corresponding author, put * next to the name then mention what the * means. * Corresponding author

Author response: Accepted and corrected accordingly in the revised manuscript.

Abstract: What is the importance of mentioning the last sentence in the introduction? Delete it and add the aim/objective of your study

Author response: last two sentences in the introduction part of the abstract conclude that both statement of the problem and the objective of the study in a precise way. Of course, we modified it by adding the objective of the study in the revised manuscript to be more clear and attractive according to the journal requirement.

Methods (page 5)

I do not think that the you have used proper study design

Author response: The study aimed to detect the incidence of early breastfeeding cessation before the WHO recommendation and time to this cessation. For this purpose, we followed birth cohorts retrospectively they were born for the last two years with certain exposure statuses mentioned as variables. We identified mothers who have children aged two to three years of age and we took their exposure status and then we followed them until the outcome variable happened (time to breastfeeding cessation or the WHO recommended time=time to event or time to censoring as operationalized in the operational definition) by recalling of the mother and the records found at the Health post. Hence, a retrospective follow-up study was used in this way. 

You have explained too much about study area which is less important, try to summarize in few sentences.

Author response: Thank you, accepted; and summarized accordingly in the revised manuscript.

What is your justification to take this timeframe October 2017 to September 20202?

Author response: This study was started at the end of September 2020. To follow mothers for two years (Follow up period), a one-year birth cohort was recruited (recruitment period). So the recruitment period started from October 1/2017 up to September 30/ 2018, and then we followed them from October 1/2018 up to September 30/2020, which was the follow-up period. That is why this time frame was taken.

The inclusion criteria is similar with the study population, try to summarize the subheadings and avoid repetition.

Author response: in the revised manuscript repetition was corrected and summarized within the single subheading as we accept your recommendation.

Exclusion criteria (last two lines), Not clear what you want to say? revise it

Author response: “Mothers who have not initially breastfed at least once for the child age two to three years now, and who gave care and breastfed other than their child” are the last two sentences in the exclusion criteria. Mothers who didn’t start breastfeeding for the child were excluded because they have no survival time (no breastfeeding duration). In the rural part of the Ethiopian context, some mothers care and breastfeed other than their own child, for example those children whose mothers were died may breastfeed the neighboring mothers breast milk and care by them. Such mothers were excluded from this study because we may not get the full survival time or censoring time during the follow-up time.

Sampling procedure: What is the total number of Kebele, from how much did you select ten? 

Author response: There are 30 kebeles in the study area and 30% of the kebeles were taken randomly. Statistically, if 20% to 30% of the population were taken using the probability sampling technique, representativeness was maintained.

Why do you use the word "infant chaild pairs" Have you collect any data from the child? I would suggest to use mothers....

Author response: For this study, participants were Mother-child pairs because in our context some mothers become pregnant and give birth even within one year of the child. In this case, the mother should be paired with the older child whose age was two to three years, otherwise, they give a response to the latest child. To avoid confusion for the data collectors and participants, mother-child pairs were used.

data collection: I do not know why you have taken date of birth as starting point? it is obvious. Your dependent variable is early BF cessation before 24 months and the participants are mothers with children 2-3 years. So why are you using retrospective follow up design? which is used to assess the outcome of certain exposure. I think the design you have used is not correct. I am ready to hear your explanation about this. The last three lines of page 8: To answer this, why do you use retrospective follow up? because it is obvious that they have started after birth then stopped in one specific time.

Author response: The dependent variable is “time to breastfeeding cessation” not only early breastfeeding cessation. The study aims to detect the survival time of breastfeeding up to WHO recommendation. So, to get the survival time the starting point of follow-up was birth (breastfeeding starts immediately after birth, nearly 30 minutes) and followed mother-child pairs until event or censoring. The exposure variables were identified during birth and then follow mother-child pairs until event or censoring. That is why a retrospective follow-up study design was used. 

Data collection tool: This should come before data collection. I would suggest to merge with the above sub headings. What was the findings of your pretest, summarize in one or two sentences? was there any problem with the tool? what do you find from your pretest?

Author response: As per your recommendation, subheadings (Data collection procedures and data collection tools) were merged as Data collection tools and procedures in the revised manuscript. Findings of pretest were summarized in the revised manuscript as “not easily understandable questionnaires’ were modified and some variables missed during pretest were incorporated in the final questionnaires”.

Results

First sub title: Why are you referring as baseline? it is confusing. do you have any data taken during follow up? I think such category is used for multiphase study but yours is single phase study. Therefore, better to sat Sociodemographic characteristics

Fourth line of first sub heading: Did you had any follow up time in your study? I this the main problem is related to the study design. What is the importance of mentioning this as incidence?

Author response: In this study, initially we took baseline maternal characteristics such as educational status, marital status, age, etc. of the mother at childbirth, and even if it is not a fully multi-phasing study, we took follow up data from the mother and other records such as immunization certificate, starting time of complementary feeding, presence of cow milk at home at least for the last two seasons. We didn’t take further sociodemographic characteristics after birth. Due to this “baseline sociodemographic characteristics” were used as a subheading. Another point you raised was “follow-up time”. It is a follow-up study from birth up to date of breastfeeding cessation or censoring time to get the survival time of breastfeeding duration. By this follow-up time, the new occurrence of breastfeeding cessation was counted and calculated the incidence rate of breastfeeding cessation before the WHO recommendation time. That is why “incidence” was written in the manuscript.

Minor comments

Title page: Be consistent, you have added affiliation to 2 and 3 but 1 and 4 did not have affiliation. Put each email next to their address

Author response: as per comments the title page consistency was revised (see the revised manuscript with track change).

Abstract: add comma to 14000, in the conclusion you have used different line spacing

data collection procedure there is a word "For 4 data collectors.... What? not clear

There are grammatical errors, the language need revision

Author response: all minor comments in the initial manuscript were modified in the revised manuscript including grammatical and language errors. For “4 data collectors” means that for interviewing mother-child pairs 2 B.Sc. nurses and 2 Health officers were involved, a total of four data collectors for this purpose. It is revised.

Reviewer #2: The draft manuscript should convince clearly with evidence the breastfeeding cessation is a critical issue in Ethiopia. Otherwise, the analyses objective should be revised to be appropriated with the country evidence-based breastfeeding issue. There is a need for the study to develop a theory of what factors and their mechanisms to predicting the breastfeeding behaviour. The arguments provided in the background is often inconsistent (See my attached written review).

Author response: Dear reviewer #2, thank you for your valuable comments and your devotion of time to this manuscript to be a full scientific paper. All scientific papers distributed to the scientific community should convince readers and be part of the solution for the problem identified in the research. This study identified that breastfeeding cessation before the WHO recommendation time was a problem that leads to 14000 childhood deaths and 5 million diarrheal and pneumonia cases annually in Ethiopia. As you explained, it is not only due to exclusive breastfeeding cessation but includes breastfeeding cessation later to 6 months that caused this problem. The proportion of Breastfeeding cessation before WHO recommended duration was less than in Ethiopia as compared to the overall sub-Saharan African countries, but the duration of breastfeeding in Ethiopia was in a decreasing trend from 2005 EDHS to 2016 EDHS (92%to 85% to 76%) with regional variation (EDHS 2016). This increased the health care costs of Ethiopia up to 0.003% GNI of the country. So that identifying whether breastfeeding cessation timing was a problem or not specifically in the rural part of Northwest Ethiopia has paramount importance. Another issue you raised is that “there is no conceptual framework of factors how that cause timing variation in breastfeeding cessation”. The original draft of this study has a conceptual framework that shows the relation between independent variables with the dependent variable, but it is not attached in the manuscript due to the journal requirement. (For further, you can see below)

Conceptual framework

6. PLOS authors have the option to publish the peer review history of their article (what does this mean?). If published, this will include your full peer review and any attached files.

Do you want your identity to be public for this peer review? For information about this choice, including consent withdrawal, please see our Privacy Policy.

Reviewer #1: No

Reviewer #2: No

Author response: No

---

## [Editor Report · Decision Letter 1]

7 Dec 2021

PONE-D-21-09209R1Time to breastfeeding cessation and its predictors among mothers who have children aged two to three years in Gozamin district, Northwest Ethiopia: a retrospective follow-up studyPLOS ONE

Dear Dr. Tilahun Degu Tsega,

Thank you for submitting your manuscript to PLOS ONE. After careful consideration, we feel that it has merit but does not fully meet PLOS ONE’s publication criteria as it currently stands. Therefore, we invite you to submit a revised version of the manuscript that addresses the points raised during the review process.

We look forward to receiving your revised manuscript.

Kind regards,

Gouranga Lal Dasvarma, PhD

Academic Editor

PLOS ONE

Journal Requirements:

Additional Editor Comments (if provided):

Thank you for addressing the reviewers' comments and suggestions. However, I cannot recommend acceptance of the manuscript yet because your marked-up copy of the revised version does not have track changes on, which makes it difficult to compare the changes you have made against the original draft. Therefore, please resubmit the revised version with track changes on. Further, please also insert the conceptual framework in the text so that I can have it reviewed.
---

## [Author Response · Author response to Decision Letter 1]

9 Dec 2021

Author Feedback: Dear editors and reviewers of this manuscript, we have no sufficient words to give our gratitude for your devotion of time and energy to improve the manuscript sufficient enough to be ready for production and then publication. As per your comments, we have tried to improve the manuscript and gave responses to those comments one by one. Dear editors and reviewers, even further we are ready to hear your comments and improve accordingly. Thanks a lot.

Editors’ comments: Thank you for addressing the reviewers' comments and suggestions. However, I cannot recommend acceptance of the manuscript yet because your marked-up copy of the revised version does not have track changes on it, which makes it difficult to compare the changes you have made against the original draft. Therefore, please resubmit the revised version with track changes on. Further, please also insert the conceptual framework in the text so that I can have it reviewed.

Authors' response: Thank you for your valuable comments and suggestions. As per your comments, we have tried to add a copy of the marked up or tracked change manuscript. Besides, we have added a conceptual framework on the manuscript with a brief text form and the full form is attached as figure -1 (Further see page 4 paragraph 2 of the manuscript).

Reviewers' comments:

Reviewer #2: The draft manuscript should convince clearly with evidence the breastfeeding cessation is a critical issue in Ethiopia. Otherwise, the analyses objective should be revised to be appropriated with the country evidence-based breastfeeding issue. There is a need for the study to develop a theory of what factors and their mechanisms to predicting the breastfeeding behaviour. The arguments provided in the background is often inconsistent (See my attached written review).

Author response: Dear reviewer #2, thank you for your valuable comments and your devotion of time to this manuscript to be a full scientific paper. All scientific papers distributed to the scientific community should convince readers and be part of the solution for the problem identified in the research. This study identified that breastfeeding cessation before the WHO recommendation time was a problem that leads to 14000 childhood deaths and 5 million diarrheal and pneumonia cases annually in Ethiopia. As you explained, it is not only due to exclusive breastfeeding cessation but includes breastfeeding cessation later to 6 months that caused this problem. The proportion of Breastfeeding cessation before WHO recommended duration was less than in Ethiopia as compared to the overall sub-Saharan African countries, but the duration of breastfeeding in Ethiopia was in a decreasing trend from 2005 EDHS to 2016 EDHS (92%to 85% to 76%) with regional variation (EDHS 2016). This increased the health care costs of Ethiopia up to 0.003% GNI of the country. So that identifying whether breastfeeding cessation timing was a problem or not specifically in the rural part of Northwest Ethiopia has paramount importance. Another issue you raised is that “there is no conceptual framework of factors how that cause timing variation in breastfeeding cessation”. The original draft of this study has a conceptual framework that shows the relation between independent variables with the dependent variable. Now, it is attached in the manuscript as per your comments. (For further, you can see at the manuscript-page 4 paragraph 2 and figure 1).

---

## [Editor Report · Decision Letter 2]

14 Dec 2021

PONE-D-21-09209R2Time to breastfeeding cessation and its predictors among mothers who have children aged two to three years in Gozamin district, Northwest Ethiopia: a retrospective follow-up studyPLOS ONE

Dear Dr. Tilahun Degu Tsega,

Thank you for submitting your manuscript to PLOS ONE. After careful consideration, we feel that it has merit but does not fully meet PLOS ONE’s publication criteria as it currently stands. Therefore, we invite you to submit a revised version of the manuscript that addresses the points raised during the review process. IN PARTICULAR, WE NEED YOU TO SUBMIT YOUR MARKED-UP COPY WITH TRACK CHANGES ON.

Please submit your revised manuscript by 21 DECEMBER 2021. If you will need more time than this to complete your revisions, please reply to this message or contact the journal office at plosone@plos.org. Please include the following items when submitting your revised manuscript:A rebuttal letter that responds to each point raised by the academic editor and reviewer(s). You should upload this letter as a separate file labelled 'Response to Reviewers'.A marked-up copy of your manuscript that highlights changes made to the original version. You should upload this as a separate file labelled 'Revised Manuscript with Track Changes'.An unmarked version of your revised paper without tracked changes. You should upload this as a separate file labeled 'Manuscript'.If applicable, we recommend that you deposit your laboratory protocols in protocols.io to enhance the reproducibility of your results. Protocols.io assigns your protocol its own identifier (DOI) so that it can be cited independently in the future. For instructions see: https://journals.plos.org/plosone/s/submission-guidelines#loc-laboratory-protocols. Additionally, PLOS ONE offers an option for publishing peer-reviewed Lab Protocol articles, which describe protocols hosted on protocols.io. Read more information on sharing protocols at https://plos.org/protocols?utm_medium=editorial-email&utm_source=authorletters&utm_campaign=protocols.

We look forward to receiving your revised manuscript.

Kind regards,

Gouranga Lal Dasvarma, PhD

Academic Editor

PLOS ONE

Journal Requirements:

Additional Editor Comments (if provided):

Please submit your marked-up copy WITH TRACK CHANGES ON.
---

## [Author Response · Author response to Decision Letter 2]

17 Dec 2021

Author Feedback: Dear editors and reviewers of this manuscript, we have no sufficient words to give our gratitude for your devotion of time and energy to improve the manuscript sufficient enough to be ready for production and then publication. As per your comments, we have tried to improve the manuscript and gave responses to those comments one by one. Dear editors and reviewers, even further we are ready to hear your comments and improve accordingly. Thanks a lot.

Editors’ comments: Thank you for addressing the reviewers' comments and suggestions. However, I cannot recommend acceptance of the manuscript yet because your marked-up copy of the revised version does not have track changes on it, which makes it difficult to compare the changes you have made against the original draft. Therefore, please resubmit the revised version with track changes on. Further, please also insert the conceptual framework in the text so that I can have it reviewed.

Authors' response: Thank you for your valuable comments and suggestions. As per your comments, we have tried to add a copy of the marked up or tracked change manuscript. Besides, we have added a conceptual framework on the manuscript with a brief text form and the full form is attached as figure -1 (Further see page 4 paragraph 2 of the manuscript).

Reviewers' comments:

Reviewer #2: The draft manuscript should convince clearly with evidence the breastfeeding cessation is a critical issue in Ethiopia. Otherwise, the analyses objective should be revised to be appropriated with the country evidence-based breastfeeding issue. There is a need for the study to develop a theory of what factors and their mechanisms to predicting the breastfeeding behaviour. The arguments provided in the background is often inconsistent (See my attached written review).

Author response: Dear reviewer #2, thank you for your valuable comments and your devotion of time to this manuscript to be a full scientific paper. All scientific papers distributed to the scientific community should convince readers and be part of the solution for the problem identified in the research. This study identified that breastfeeding cessation before the WHO recommendation time was a problem that leads to 14000 childhood deaths and 5 million diarrheal and pneumonia cases annually in Ethiopia. As you explained, it is not only due to exclusive breastfeeding cessation but includes breastfeeding cessation later to 6 months that caused this problem. The proportion of Breastfeeding cessation before WHO recommended duration was less than in Ethiopia as compared to the overall sub-Saharan African countries, but the duration of breastfeeding in Ethiopia was in a decreasing trend from 2005 EDHS to 2016 EDHS (92%to 85% to 76%) with regional variation (EDHS 2016). This increased the health care costs of Ethiopia up to 0.003% GNI of the country. So that identifying whether breastfeeding cessation timing was a problem or not specifically in the rural part of Northwest Ethiopia has paramount importance. Another issue you raised is that “there is no conceptual framework of factors how that cause timing variation in breastfeeding cessation”. The original draft of this study has a conceptual framework that shows the relation between independent variables with the dependent variable. Now, it is attached in the manuscript as per your comments. (For further, you can see at the manuscript-page 4 paragraph 2 and figure 1).

---

## [Editor Report · Decision Letter 3]

26 Dec 2021

PONE-D-21-09209R3Time to breastfeeding cessation and its predictors among mothers who have children aged two to three years in Gozamin district, Northwest Ethiopia: a retrospective follow-up studyPLOS ONE

Dear Dr. Tilahun Degu Tsega, MPH

Thank you for submitting your manuscript to PLOS ONE. After careful consideration, we feel that it has merit but does not fully meet PLOS ONE’s publication criteria as it currently stands. Therefore, we invite you to submit a revised version of the manuscript that addresses the points raised during the editorial review process. Thank you for uploading the revised manuscript with track changes on, which has made it possible for me to compare your responses with the reviewers' comments. However, minor editorial omissions continue to remain in Revision#3. For example, in the Introduction to the Abstract you have cited the number 14000 without a comma as a ,000 separator. Please write the number as 14,000. Similarly, in Methods in the Abstract you have started the sentence with a lower case "a" such as "a community-based retrospective follow-up study was used...." . This should be written as "A community-based retrospective follow-up study was used....". (i.e. beginning with an upper case "A"). PLOS One does not provide copy editing, therefore please make the corrections as indicated above, and have the entire revised manuscript thoroughly edited for English.

We look forward to receiving your revised manuscript.

Kind regards,

Gouranga Lal Dasvarma, PhD

Academic Editor

PLOS ONE
---

## [Author Response · Author response to Decision Letter 3]

28 Dec 2021

Response to Reviewers

Author Feedback: Dear editors and reviewers of this manuscript, we have no sufficient words to give our gratitude for your devotion of time and energy to improve the manuscript sufficient enough to be ready for production and then publication. As per your comments, we have tried to improve the manuscript and gave responses to those comments one by one. Dear editors and reviewers, even further we are ready to hear your comments and improve accordingly. Thanks a lot.

Editors’ comments: Thank you for uploading the revised manuscript with track changes on, which has made it possible for me to compare your responses with the reviewers' comments. However, minor editorial omissions continue to remain in Revision#3. For example, in the Introduction to the Abstract you have cited the number 14000 without a comma as a ,000 separator. Please write the number as 14,000. Similarly, in Methods in the Abstract you have started the sentence with a lower case "a" such as "a community-based retrospective follow-up study was used...." . This should be written as "A community-based retrospective follow-up study was used....". (i.e. beginning with an upper case "A"). PLOS One does not provide copy editing, therefore please make the corrections as indicated above, and have the entire revised manuscript thoroughly edited for English.

Authors' response: Thank you for your valuable comments and suggestions. As per your comments, we have tried to improve typological and grammatical problems on the revised manuscript.

---

## [Editor Report · Decision Letter 4]

30 Dec 2021

Time to breastfeeding cessation and its predictors among mothers who have children aged two to three years in Gozamin district, Northwest Ethiopia: a retrospective follow-up study

PONE-D-21-09209R4

Dear Tilahun Degu Tsega

We’re pleased to inform you that your manuscript has been judged scientifically suitable for publication and will be formally accepted for publication once it meets all outstanding technical requirements.

Kind regards,

Gouranga Lal Dasvarma, PhD

Academic Editor

PLOS ONE
---

## [Editor Report · Acceptance letter]

11 Jan 2022

PONE-D-21-09209R4 

Time to breastfeeding cessation and its predictors among mothers who have children aged two to three years in Gozamin district, Northwest Ethiopia: a retrospective follow-up study 

Dear Dr. Tsega:

I'm pleased to inform you that your manuscript has been deemed suitable for publication in PLOS ONE. Congratulations! Your manuscript is now with our production department. 

Kind regards, 

on behalf of

Dr. Gouranga Lal Dasvarma 

Academic Editor

PLOS ONE